# Chronic Ouabain Targets Pore-Forming Claudin-2 and Ameliorates Radiation-Induced Damage to the Rat Intestinal Tissue Barrier

**DOI:** 10.3390/ijms25010278

**Published:** 2023-12-24

**Authors:** Alexander G. Markov, Alexandra A. Livanova, Arina A. Fedorova, Violetta V. Kravtsova, Igor I. Krivoi

**Affiliations:** Department of General Physiology, St. Petersburg State University, 199034 St. Petersburg, Russia; alexandralivanova@mail.ru (A.A.L.); a.fedorova@spbu.ru (A.A.F.); violettakravtsova@gmail.com (V.V.K.); iikrivoi@gmail.com (I.I.K.)

**Keywords:** ionizing radiation, ouabain, epithelium, tight junctions, claudins, gastrointestinal acute radiation syndrome

## Abstract

Ionizing radiation (IR) causes disturbances in the functions of the gastrointestinal tract. Given the therapeutic potential of ouabain, a specific ligand of the Na,K-ATPase, we tested its ability to protect against IR-induced disturbances in the barrier and transport properties of the jejunum and colon of rats. Male Wistar rats were subjected to 6-day intraperitoneal injections of vehicle or ouabain (1 µg/kg/day). On the fourth day of injections, rats were exposed to total-body X-ray irradiation (10 Gy) or a sham irradiation. Isolated tissues were examined 72 h post-irradiation. Electrophysiological characteristics and paracellular permeability for sodium fluorescein were measured in an Ussing chamber. Histological analysis and Western blotting were also performed. In the jejunum tissue, ouabain exposure did not prevent disturbances in transepithelial resistance, paracellular permeability, histological characteristics, as well as changes in the expression of claudin-1, -3, -4, tricellulin, and caspase-3 induced by IR. However, ouabain prevented overexpression of occludin and the pore-forming claudin-2. In the colon tissue, ouabain prevented electrophysiological disturbances and claudin-2 overexpression. These observations may reveal a mechanism by which circulating ouabain maintains tight junction integrity under IR-induced intestinal dysfunction.

## 1. Introduction

Ionizing radiation (IR) is a unique physical factor that affects all tissues and organ systems, including the gastrointestinal tract [1]. IR damage to the gut can develop during total-body irradiation as a result of nuclear power plant incidents and radiation accidents, as a complication after radiation therapy in cancer patients [2,3], during space flights (0.3–0.6 mGy/day) [4]. Intraoperative radiation therapy dose is generally in the range of 10–20 Gy [5,6]. A similar dose load is impacted on personnel by nuclear power plant incidents and radiation accidents; for example, Chernobyl nuclear plant workers received doses up to 16 Gy [7]. After exposure at doses exceeding the 6 Gy threshold, mammals develop gastrointestinal acute radiation syndrome (GIARS) characterized by gastrointestinal leisure due to epithelial barrier damage [8,9].

The intestinal epithelium is a multifunctional tissue barrier that provides transport of ions and macromolecules, as well as protection against the penetration of microorganisms from the lumen into the internal environment of the body. Tight junctions (TJs) unite the lateral surfaces of epithelial cells of the intestinal mucosa [10]. They provide integrity of the epithelium with selective intercellular transport of ions, water, and organic macromolecules [11].

The most important structural and functional molecular determinants of the TJs are claudins, a superfamily of integral proteins that form the TJs and determine paracellular permeability and barrier properties of epithelium, endothelium, and mesothelium [12,13,14]. Cells in these tissues co-express different claudins, which differentially regulate TJs selectivity and permeability, depending on the type of claudin expressed. Claudin-1, -3, and -4 are recognized as barrier-forming proteins, playing a crucial role in diminishing the permeability of the paracellular barrier. In contrast, claudin-2 is a pore-forming one that increases permeability to cations, mainly Na^+^ and water molecules [15,16,17] and is a protein of particular interest. Intestinal barrier dysfunction due to inflammation [18] or cholera toxin [19] is associated with increased claudin-2 abundance (for review [20]). In addition to claudins, TJs include proteins of the TAMP (Tight junction–Associated MARVEL Proteins) family, in particular occludin and tricellulin, which regulate the permeability of the intestinal barrier to macromolecules [21,22].

Experiments with different doses of total-body X-ray IR (nonlethal—2 Gy, half-lethal—5 Gy, and lethal—10 Gy ranges) showed that only exposure to a dose of 10 Gy led to the manifestation of physiological effects on the barrier functions of the rat jejunum and colon 72 h after IR [23,24]. Previous findings demonstrated that IR affects the expression and distribution of TJ proteins in the mammalian gut [24,25,26,27,28]. Thus, multi-dimensional modulation of claudin-2, -3, and -4 expressions in the ileac epithelium of rats was observed after abdominal 12-Gy irradiation [25]. After IR at a dose lower than the range causing GIARS, a decrease of occludin, claudin-3, and ZO-1 was shown in the murine ileum and colon [26]. Interestingly, certain parts of the rat intestine express the claudin mosaic in a segment-dependent manner, which correlates with their functional properties [29], and the jejunum and colon demonstrated different sensitivity to IR [24]. This means that electrophysiological parameters, permeability, changes in histological structure, and expression of TJ proteins in the colon and small intestine differ when exposed to the same dose of 10 Gy. In the jejunum, ileum, and colon of a nonhuman primate model, 6.7 and 7.4 Gy IR induced segment-specific and time-dependent changes in the expression of TJ-associated proteins, including claudin-2 [27].

Na,K-ATPase is a ubiquitous transport protein that pumps Na^+^ and K^+^ ions across the plasma membrane, playing a vital role in all living cells. In epithelial cells, Na,K-ATPase plays a critical role in polarity and vectorial transport [30]. In addition, a number of studies have demonstrated the involvement of Na,K-ATPase in the formation and regulation of TJ structure and permeability through modulation of claudin expression [31,32,33,34,35], suggesting a functional interaction between these proteins. Ouabain, a specific ligand for the Na,K-ATPase, when it occupies a binding site in the α subunit of Na,K-ATPase, triggers this interaction.

Ouabain is a cardiotonic steroid originally isolated from plants, and its endogenous structural analog has been demonstrated to circulate in the subnanomolar concentration range [36,37]. Endogenous ouabain is now described as a hormone that is synthesized and secreted by the adrenal glands and hypothalamus [37], and its unique role in health and disease is well documented. Ouabain has broad therapeutic potential and is thought to be involved in salt handling, inflammation, neuroprotection, neural signaling and behavior, gene expression, cell growth, and differentiation [37,38,39,40,41] while also exhibiting anticancer [42] and antiviral activity [43].

Previous in vitro studies on cultured cells and in vivo experiments have shown that nanomolar concentrations of ouabain modulate both tightening claudin-1, -4 and pore-forming claudin-2 expression [31,32,33,34], suggesting a physiological mechanism for the regulation of epithelial phenotype by circulating ouabain. Chronic elevation of circulating ouabain by injections has been shown to protect against lipopolysaccharide-induced epithelial damage in the lungs of mice [44] as well as the intestines of rats [34]. IR induces structural and functional damage to cellular membranes and membrane-bound enzymes and disrupts the Na,K-ATPase function [45,46,47]. In skeletal muscle, chronic administration of ouabain prevented such damage [47].

Investigating the molecular mechanisms underlying IR-induced intestinal injury and identifying countermeasures is of great importance and remains challenging. The claudins listed above have been noted as potential targets for therapy in intestinal diseases [48]. Previously, in animal models of various pathological conditions, injections of ouabain in doses of 1–1.8 μg/kg demonstrated their effectiveness [34,47,49,50]. In summary, we hypothesized that elevations of circulating ouabain by its exogenous administration could maintain intestinal barrier function during IR-induced injury. We tested this hypothesis in rats intraperitoneally injected with ouabain in a dose of 1 μg/kg and consequently exposed to total-body X-ray IR (10 Gy). In isolated tissues of the jejunum and colon, electrophysiological characteristics and paracellular permeability were studied; Western blotting and histological analysis were also performed.

## 2. Results

### 2.1. Chronic Ouabain Elevates the Serum Ouabain Level and Prevents IR-Induced Electrophysiological Disturbances in Rat Colon but Not Jejunum Tissue

The general design of the study is presented in Figure 1a–d. In control rats, the concentration of ouabain in the blood serum was 0.7 ± 0.1 nM, and IR significantly (*p* < 0.05) increased this concentration to 2.5 ± 0.7 nM. Chronic administration of ouabain significantly increased serum ouabain concentration to 1.5 ± 0.3 nM (*p* < 0.05), and in IR-exposed ouabain-treated rats, ouabain concentration increased (*p* < 0.05) to 17.3 ± 5.8 nM (Figure 1a–d).

Transepithelial resistance (TER) was evaluated in an Ussing chamber as a physiological indicator of barrier and transport functions of jejunum and colon tissues. In the jejunum tissue, IR significantly reduced TER fivefold (*p* < 0.001) compared to the control. Chronic treatment with ouabain itself did not affect TER and did not prevent IR-induced TER impairment (Figure 2a). In the colon tissue, IR also significantly (*p* < 0.001) decreased TER compared to control. However, although chronic ouabain alone was ineffective, it prevented the TER impairment caused by IR (Figure 2c). 

Thus, chronic ouabain demonstrated a segment-specific protective effect preventing IR-induced disturbances in electrophysiological parameters of rat colon but not jejunum tissue.

Paracellular permeability was measured in an Ussing chamber as sodium fluorescein (376 Da) flux. In the jejunum tissue, exposure to IR resulted in a twofold increase (*p* < 0.05) in paracellular permeability compared to control. Chronic ouabain treatment was ineffective either alone or in combination with IR (Figure 2b). Paracellular permeability in the colon tissue showed complete resistance to both ouabain, IR, and their combination (Figure 2d).

### 2.2. Chronic Ouabain Does Not Prevent IR-Induced Morphological Disruptions in Rat Jejunum and Colon Tissues

In the jejunum tissue, IR caused a shortening of the villi and an increase in their diameter. In addition, an increase in the submucosal layer and a decrease in the depth of the crypts without changing their diameter were observed. The muscular and adventitial layers did not change. Chronic ouabain increased crypt depth compared to control; other parameters were not changed. Finally, chronic ouabain was unable to prevent IR-induced disturbances in the histological characteristics of jejunum tissue (Figure 3).

In the colon tissue, IR caused only a decrease (*p* < 0.001) in crypt depth, as well as an increase in the muscle and adventitial layers (*p* < 0.05). Chronic ouabain by itself did not affect any parameters and could not prevent the observed disturbances (Figure 4).

### 2.3. Chronic Ouabain Prevents IR-Induced Claudin-2 Upregulation in Rat Jejunum and Colon Tissues

In the jejunum tissue, IR significantly (*p* < 0.05) increased the levels of claudin-1 and -2 compared to control. In addition, claudin-3 and -4 were identified after IR, although they were not detected in the control. IR also caused an increase in occludin levels, whereas tricellulin and activated caspase-3 levels decreased. Chronic ouabain itself did not affect the levels of the studied proteins and did not prevent IR-induced changes in their levels, with the exception of claudin-2 and occludin. Chronic ouabain prevented the IR-induced upregulation of these two proteins, and their levels were not significantly different from the control (Figure 5a,b).

Colon tissue showed resistance to IR, and only claudin-2 and -4 were upregulated (*p* < 0.05). Chronic ouabain alone did not affect the levels of the proteins studied but specifically prevented IR-induced upregulation of claudin-2 (Figure 5c,d).

## 3. Discussion

Previous studies have shown multi-modal negative effects of IR on biological systems, including the gastrointestinal tract. Among them, disturbances in claudin expression, TJs integrity, and epithelial barrier properties are of vital importance [24,25,26,27].

The mechanisms by which IR affects gene expression are complex and multifaceted. IR directly interacts with DNA molecules, causing various types of DNA damage, such as single-strand breaks, double-strand breaks, and base damage [51,52]. It is known that IR causes the formation of free radicals and the production of reactive oxygen species (ROS), depletion of the antioxidant status of cells, and their death [53]. Genes involved in antioxidant defense mechanisms, such as superoxide dismutase and catalase, may be induced in response to radiation-induced oxidative stress [54].

Inflammation is also one of the serious abnormalities caused by IR [55]. In addition, IR potentiates lipid peroxidation, which disorganizes the arrangement of the lipid bilayer [56,57]. Protection from the harmful effects of IR is a high priority, and various strategies are being studied, including improving the activity of the antioxidant system [26,58,59]. In particular, it has been shown that N-acetyl-L-cysteine can serve as a potential radioprotector against IR-induced intestinal damage in mice [26]. Since nuclear factor kappa B (NF-κB) is involved in the regulation of apoptosis, as well as immune and inflammatory responses, it may also be a potential target for mitigating the IR response [60,61].

Na,K-ATPase is a ubiquitous transport protein that pumps Na^+^ out of the cell and K^+^ in the opposite direction across the plasma membrane. The catalytic and transport α subunit of Na,K-ATPase is expressed in four isoforms in a cell- and tissue-specific manner. Most cells co-express the α1 isoform in combination with other α isoforms, whereas red blood cells, kidneys, lungs, and intestine predominantly express the α1 isoform [62,63]. The extracellular loops of the α subunit form a highly specific binding site for cardiotonic steroids such as ouabain and digoxin, and this site is the only known receptor for these ligands [64]. Ouabain is a cardiotonic steroid-type compound. Ouabain, originally extracted from plants (e.g., *Strophanthus gratus* and *Acokanthera ouabaio*), has an endogenous circulating analog that has been isolated from mammalian tissues and fluids and is structurally, biochemically, and immunologically indistinguishable from exogenous ouabain [65]. Ouabain is believed to be synthesized in the adrenal gland cortex and hypothalamus and is currently suggested to act as a hormone [37].

Both exogenous and endogenous ouabain are now recognized as multimodal cell modulators with broad therapeutic potential [37,38,39,40,41,42,43]. Endogenous ouabain normally circulates in the subnanomolar concentration range, but its levels can vary significantly in a number of physiological and pathophysiological conditions [37,38,40,66,67,68]. In animal models of inflammation, low concentrations of ouabain have been shown to attenuate oxidative stress and lipid peroxidation impairment [50] and also protect against NF-κB activation [41,44].

The ability of nanomolar (10 nM) concentrations of ouabain to affect claudin-1, -2, and -4 expression and epithelial barrier properties was first demonstrated in in vitro studies [31,32,33]. In vivo studies [34] confirmed the functional Na,K-ATPase/claudin interaction and the importance of ouabain as a circulating regulator of this interaction. Ouabain, when it occupies a specific binding site in the α1 subunit of Na,K-ATPase, triggers Src/Erk1/2-dependent intracellular signaling that regulates claudin expression, epithelial phenotype, and proliferation [31,32,33,34]. However, physiological levels of ouabain (3 nM) can also promote apoptosis and cell death, as has been demonstrated in renal epithelial cells [69]. 

Notably, IR increases lipid peroxidation and affects plasma membrane lipids, leading to changes in membrane integrity and fluidity [56,57]. Concomitant structural disintegration of the plasma membrane can be accompanied by functional impairments of integral proteins, including Na,K-ATPase [70], and claudins. Thus, in addition to ROS production and DNA damage, IR may affect the functional interaction of Na,K-ATPase/claudins interaction through the destruction of plasma membrane lipids.

To enhance the concentration of circulating ouabain, intraperitoneal injections of ouabain at doses of 1–1.8 µg/kg are widely used [49,50,68,71]. In our study, we found that chronic administration of exogenous ouabain (1 μg/kg) increased circulating ouabain levels approximately twofold (from 0.7 nM in control to 1.5 nM). Interestingly, IR also increased serum ouabain concentrations to 2.5 nM, suggesting that IR itself may stimulate the synthesis and release of endogenous ouabain. In rats chronically treated with exogenous ouabain and exposed to IR, the serum ouabain concentration increased dramatically to 17.3 nM.

The mechanisms of these changes in serum ouabain concentrations are unclear. First, the form in which ouabain circulates is poorly understood. It is proposed that ouabain is transported in complexes with carrier(s) proteins that provide a reservoir/buffer for ouabain and protection from metabolism and renal clearance. It is also impossible to exclude some feedback mechanisms that regulate the dissociation of ouabain from the carrier and, consequently, the level of its circulation [72]. In this study, six days of exogenous ouabain administration increased serum ouabain levels. Studies in dogs and humans have shown that following a single intravenous injection, plasma ouabain concentration, after an initial rapid decline, further declines slowly with a half-life of 18–24 h. Repeated daily administration of ouabain to humans resulted in plateau plasma concentrations after 4–5 days, suggesting equilibrium between ouabain accumulation and clearance [73]. How IR may influence these mechanisms has not been studied. Secondly, the synthesis and release of endogenous ouabain are regulated by the sympathetic nervous system [37,74], and the hypothalamic-pituitary-adrenal axis is known to be activated by IR [75]. Thus, an increase in ouabain levels in rats receiving exogenous ouabain and exposed to IR may further activate the sympathetic nervous system, which may additionally stimulate the release of ouabain from the adrenal glands.

The main finding of this study is that chronic administration of ouabain prevented IR-induced overexpression of claudin-2. Notably, pore-forming claudin-2, which increases permeability for cations, mainly Na^+^ and water molecules [15,16,17,20], is a protein of particular interest. The role of claudin-2 in the colon depends on the agent or conditions that caused its upregulation. In the case of bacterial infection or toxins, an increase in claudin-2 levels is considered an adaptive process aimed at flushing out the toxin or bacteria [20]. In inflammation of the colon, such as Crohn’s disease, an increase in claudin-2 is considered to be a pathogenetic change in the state of the intestinal barrier [76]. Upregulation of claudin-2 has been described in cholera toxin exposure [19], inflammation [18,77], and several other pathologies, as well as IR exposure in this and other studies [24,25,27].

IR induces the overproduction of ROS, which is a multifaceted regulator that plays an important role in various pathways involved in maintaining cellular homeostasis and regulating key transcription factors [78]. The factors that mediate claudin-2 upregulation are not fully characterized, but various studies have shown that tumor necrosis factor (TNF) and a number of interleukins are potential enhancers of claudin-2 expression [20]. IR is known to enhance the production of TNF-α and interleukins [28,79]. In particular, total body irradiation at a dose of 10 Gy increases serum levels of TNF-α and IL-6 in humans [5], as well as in human endothelial cells [80]. It has been suggested that TNFα-dependent upregulation of claudin-2 may occur through the phosphatidylinositol-3-kinase pathway [81]. The role of TNF-α/NF-kB signaling in the upregulation of claudin-2 has also been reported [82,83]. The activity of the claudin-2 promoter has been shown to be increased by IL-6 in a MEK/ERK and PI3K-dependent manner, resulting in increased expression of claudin-2 [84].

It is noteworthy that IR (8.5 Gy) at 3.5 days after irradiation changed the localization of claudin-2 in epithelial cells of the mice’s small intestine [28]. It has also been shown that increased expression of claudin-2 is associated with increased binding of claudin-2 and caveolin-1 [85]. Caveolin-1 is a vital protein for many cellular processes, including autophagy [86]. Thus, increased binding of caveolin-1 and claudin-2 suggests that caveolin-1 may act as a shuttle mechanism for increased internalization of claudin-2 [85]. 

Notably, ouabain is able to activate AMP-activated protein kinase (AMPK), a master regulator of cellular metabolism in eukaryotes that plays a critical role in cellular processes such as autophagy [87]. Moreover, ouabain at a concentration of 25 nM (which is similar to the 17.3 nM observed in this study) activates AMPK and positively regulates autophagy in human cancer cell lines [88]. Thus, this autophagic degradation pathway triggered by ouabain may further enhance caveolin-1-dependent claudin-2 internalization. If so, such a mechanism could explain the preventive effect of ouabain administration against IR-induced claudin-2 upregulation. Combining the literature and our findings, we propose the following hypothetical scheme, presented in Figure 6.

Importantly, the cellular cytoskeleton, composed of a dynamic network of microfilaments, intermediate filaments, and microtubules, plays an important role in cell polarity, barrier integrity, intracellular trafficking, and intestinal epithelium homeostasis [89]. The ROCK (Rho-associated coiled-coil forming kinase) signaling pathway is considered a key regulator of the cytoskeleton components [90]. In cultured epithelial (MDCK) cells, ouabain (10 nM) has been shown to induce transcript changes and activation of ROCK signaling [91], and this finding opens a new field of further research on claudin-2 regulation. Whether occludin, which is associated with the actin cytoskeleton and regulated by the ROCK signaling pathway [92], may be involved in the regulation of claudin-2 expression triggered by ouabain remains to be determined.

It should be noted that in rodents, the α1 Na,K-ATPase isozyme is highly resistant to ouabain (IC_50_ values range from tens to hundreds micromolar), compared to other isozymes that are two to four orders of magnitude more sensitive [62,64]. This suggests that the circulating ouabain concentrations in our study trigger α1Na,K-ATPase/Src-dependent intracellular signaling rather than inhibiting enzyme activity and altering Na^+^ balance.

However, it is important to note that, unlike in rodents, the α1 isozyme in rabbits, pigs, dogs, sheep, guinea pigs, and humans is relatively sensitive to ouabain. In humans, a similar affinity of all α subunit isozymes with ouabain binding constants in the nanomolar concentration range has been shown [62,64,93,94]. This leaves open the question of how far the results of our study can be extrapolated to subjects other than rats, and this may be considered a limitation of this work. Altogether, our new findings corroborate a functional α1Na,K-ATPase/claudin interaction and the importance of ouabain as a circulating regulator of this interaction that can modulate claudin-2 expression. Further studies of dose-, time-, and use-dependence are needed to more accurately evaluate the therapeutic potential of ouabain treatment for IR-induced intestinal injury. 

## 4. Materials and Methods

### 4.1. Animals and Experimental Design

The study involved male Wistar rats (10 weeks; 230–250 g) maintained in a controlled environment with ad libitum access to food and water. All animal experiments were performed in accordance with the Guide for the Care and Use of Laboratory Animals [95]. The experimental protocol was in accordance with the requirements of the EU Directive 2010/63/EU for animal experiments and approved by the Bioethics Committee of St. Petersburg State University no. 131-03-5 (issued 13-12-2017).

The study design is presented in Figure 1a–d. Animals (*n* = 26) were randomly divided into four groups: (a) ‘Control’ (*n* = 6)–were subjected to intraperitoneal injections of 1 mL sterile 0.9% NaCl (vehicle) for 6 days, on the fourth day were subjected to sham irradiation procedure; (b) ‘Radiation’ (*n* = 7)–were subjected to injections of 1 mL sterile 0.9% NaCl (vehicle) for 6 days, on the fourth day were exposed to 10-Gy total-body X-ray irradiation; (c) ‘Ouabain’ (*n* = 6)–were subjected to intraperitoneal injections of ouabain at a dose of 1 µg/kg for 6 days, on the fourth day were subjected to sham irradiation procedure; (d) ‘Ouabain + Radiation’ (*n* = 7)–were subjected to injections of ouabain at a dose of 1 μg/kg for 6 days, on the fourth day were exposed to 10-Gy total-body X-ray irradiation.

Rats were euthanized 72 h after exposure to IR (on the seventh day of the experiment) by the intraperitoneal injection of a tribromoethanol overdose (750 mg/kg) followed by decapitation. The jejunum and colon were isolated, and mixed blood was collected. Freshly isolated fragments of the jejunum and colon were immediately used for electrophysiological measurements. Other fragments were collected and then stored either at −80 °C for later Western blot analysis or in 10% formalin for histological assessment.

### 4.2. Exposure to X-ray Ionizing Radiation

Rats were exposed to single total-body X-ray irradiation (10 Gy) using the RUM-17 orthovoltage therapeutic X-ray unit (MosRentgen, Moscow, Russia). During the irradiation procedure, the animals were placed in a closed Plexiglas box, which completely restricted their movement. The focal length of the X-ray tube was 50 cm; the dose rate was 0.31 Gy/min. This means that if an animal is under the switched-on X-ray tube for a minute, it receives an absorbed dose of 0.31 Gy. Accordingly, after 32 min it receives 10 Gy. To check the absorbed dose, an individual dosimeter was used, followed by result interpretation with a GO-32 measuring device (Spetsoborona, Saint Petersburg, Russia). For the procedure of sham irradiation, the animals were placed in a box under the turned-off X-ray tube for the same exposure time. 

### 4.3. Biochemical Analysis of Ouabain Concentration in Blood Samples

Serum concentrations of ouabain were measured with the ELISA Kit for ouabain (CEV857Ge, Cloud-Clone Corp., Katy, TX, USA). The assay’s procedures were conducted in accordance with the manufacturer’s protocols, and the light absorbance was measured at 450 nm with a spectrophotometric microplate reader SPECTROstar Nano (BMG Labtech, Ortenberg, Germany).

### 4.4. Registration of Electrophysiological Parameters in the Ussing Chamber

Barrier properties of the colon epithelium (transepithelial resistance, TER) were assessed in the Ussing chamber, following established procedures [23]. During 60-min registration, jejunum and colon fragments (0.13 cm^2^ exposed area) were maintained in a solution with defined ionic composition (in mM: NaCl, 119; KCl, 5; CaCl_2_, 1.2; MgCl_2_, 1.2; NaHCO_3_, 25; Na_2_HPO_4_, 1.6; NaH_2_PO_4_, 0.4; d-glucose, 10; pH 7.4) at 37 °C. TER was calculated using Ohm’s law. Data were averaged from 3 to 4 jejunum and colon fragments per rat.

### 4.5. Permeability for Sodium Fluorescein

Fragments of the colon and jejunum were mounted in Ussing chambers as described earlier. 50 μL of 376 Da sodium fluorescein (Sigma Aldrich, Burlington, MA, USA) was placed in the Ussing chamber from the apical side to obtain a final concentration of 0.1 mM. The solution from the basolateral side was collected after 60 min of incubation to determine the sodium fluorescein diffused through the tissue. The signal intensity was measured using a Typhoon^TM^ FLA 9500 laser scanner (GE Healthcare, Chicago, IL, USA) using an excitation wave with a length of 473 nm and a voltage of 430 V. The obtained images were analyzed using ImageJ version 1.52a software (National Institutes of Health, Bethesda, MD, USA). The permeability value (Papp) was calculated using the formula Papp = (dQ/qt)/(A × C0), where (dQ/qt) is the concentration of sodium fluorescein on the serous side of the Ussing chamber after 60 min of incubation (M); A—area of the tissue area under study (cm^2^); C0 is the concentration of sodium fluorescein in solution from the mucosal side at the initial time (M). Considering the relation 1 L = 1000 cm^3^, the permeability dimension is presented in cm/s.

### 4.6. Histological Analysis

Fragments of the jejunum and colon were isolated from all groups of animals and placed in a solution of 10% formalin (BioVitrum, St. Petersburg, Russia) for 48 h. The procedure for embedding tissues in paraffin blocks was carried out according to a routine technique. Using a Leica RM2265 rotary microtome (Leica Microsystems, Wetzlar, Germany), 5-μm-thick tissue sections were prepared, mounted on glass slides, and used for further morphological examination by light microscopy. Sections were stained with hematoxylin (1 min, 23 °C) and eosin (30 s, 23 °C) (H&E) and studied using a Leica DMI6000 light microscope (Leica Microsystems, Wetzlar, Germany). Microphotographs were taken with a color digital camera and analyzed using Leica Application Suite software 5.1.0 (Leica Microsystems, Wetzlar, Germany).

To assess changes in the histological structure of the tissues of the jejunum and colon after irradiation, quantitative morphometry of the obtained images was performed using ImageJ software (National Institutes of Health, Bethesda, MD, USA). For each animal, two tissue fragments were taken, with at least two sections per fragment, and five regions of interest per slide were examined. Quantitative analysis was performed blindly.

### 4.7. Western Blotting

To determine the level of proteins of the claudin family, occludin, tricellulin, and activated caspase-3, the Western blotting in Stain-Free^TM^ modification was performed. Fragments of tissue from the jejunum and colon were subjected to mechanical homogenization and lysation at RIPA buffer (in mM: Tris, 20; NaCl, 150; Triton X-100, 1; EDTA, 5; tween-20, 0.1; protease inhibitors cOmplete mini tablets (Roche, Mannheim, Germany, pH 7.6). Samples were incubated on ice for 30 min, then sonicated and centrifuged for 15 min at 13,000× *g* at 4°C (Eppendorf, Hamburg, Germany). Total protein contents in collected supernatants were measured with a Pierce Rapid Gold Bicinchoninic Acid Protein Assay Kit (Thermo Fisher Scientific, Waltham, MA, USA), according to the manufacturer’s protocol, using a spectrophotometric microplate reader SPECTROstar Nano (BMG Labtech, Ortenberg, Germany). Equal amounts of total protein were heated for 10 min at 95 °C or 37 °C (for tricellulin assay only) with a 4× Laemmli buffer.

Vertical Stain Free^TM^ gel electrophoresis was performed to separate proteins by molecular weight. Semi-dry electrotransfer of protein content to PVDF membranes (Bio-Rad, Hercules, CA, USA) was carried out at a constant voltage of 25 V. After blocking for 2 h with 5% skimmed milk at room temperature, PVDF membranes were incubated overnight at 4 °C with 2.5% skimmed milk and either primary mouse antibody (against claudin-2 and occludin, #32-5600, #33-1500, Thermo Fisher Scientific, Waltham, MA, USA) or primary rabbit antibody (against claudin-1, -3, -4, tricellulin, #71-7800, #34-1700, #36-45800, #48-8400, Thermo Fisher Scientific, Waltham, MA, USA; against activated caspase-3, #9661s, CellSignaling, MA, USA). The next day, after intensive washing, the membranes were incubated for 45 min with 2.5% skimmed milk and horseradish-peroxidase-conjugated anti-mouse (#AB205719, Abcam, Eugene, OR, USA) or anti-rabbit secondary antibody (#a3273, Abcam, Eugene, OR, USA). Detection of the chemiluminescent signal was carried out on a Chemi-Doc XRS+ image analyzer (Bio-Rad, Hercules, CA, USA). Target band signal densities were normalized to total protein. The level of protein in the control group of animals was considered as 100%, and the values in other groups were calculated relatively to control. In the absence of a signal from the corresponding proteins, the densitometry signal was considered as zero. Due to the lack of signal for claudin-3 and -4 in the small intestine, the signal density in the ‘Radiation’ group was taken as 100%.

### 4.8. Statistics

Statistical analysis was performed using GraphPad Prism 8 software (GraphPad, San Diego, CA, USA). The difference between groups was estimated using one-way ANOVA followed by Bonferroni multiple comparisons test. All data are presented as mean ± standard error of the mean.

## Figures and Tables

**Figure 1 ijms-25-00278-f001:**
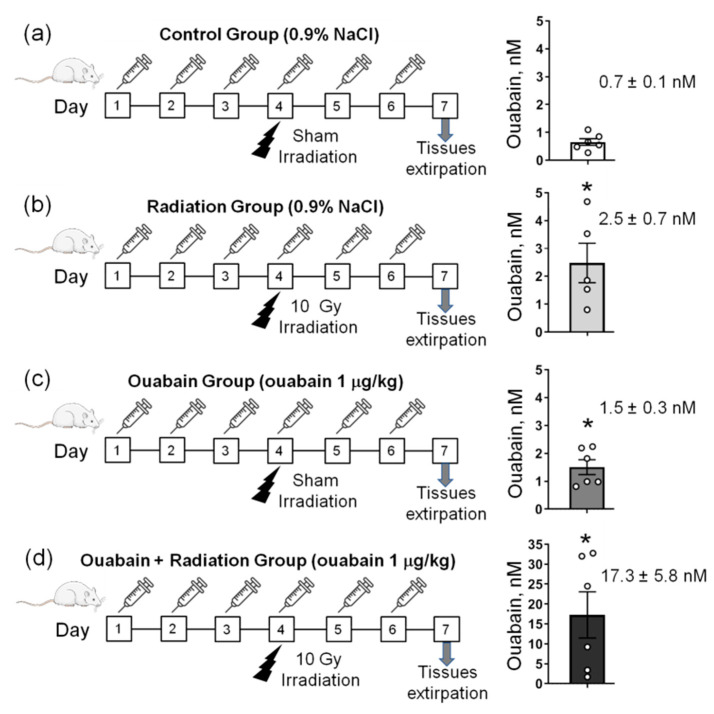
The study design. (**a**–**d**) Rats were subjected to daily intraperitoneal injections of either 1 mL sterile 0.9% NaCl (vehicle)—(**a**,**b**) or ouabain (1 µg/kg body weight) dissolved in 1 mL sterile 0.9% NaCl (**c**,**d**). On the fourth day of injections, rats were exposed to sham-irradiation procedure (**a**,**c**) and 10 Gy irradiation (**b**,**d**). Thus, there were four experimental groups: rats subjected to injections of saline without exposure to irradiation—‘Control’ group (**a**); vehicle-injected rats exposed to total-body X-ray irradiation (10 Gy)—‘Radiation’ group (**b**); rats treated with ouabain without exposure to irradiation—‘Ouabain’ group (**c**); and rats treated with ouabain and exposed to irradiation—‘Ouabain + Radiation’ group (**d**). The jejunum and colon were dissected 72 h after exposure to irradiation. Right–corresponding serum ouabain levels. The number of rats corresponds to the number of symbols. * *p* < 0.05—compared to the control.

**Figure 2 ijms-25-00278-f002:**
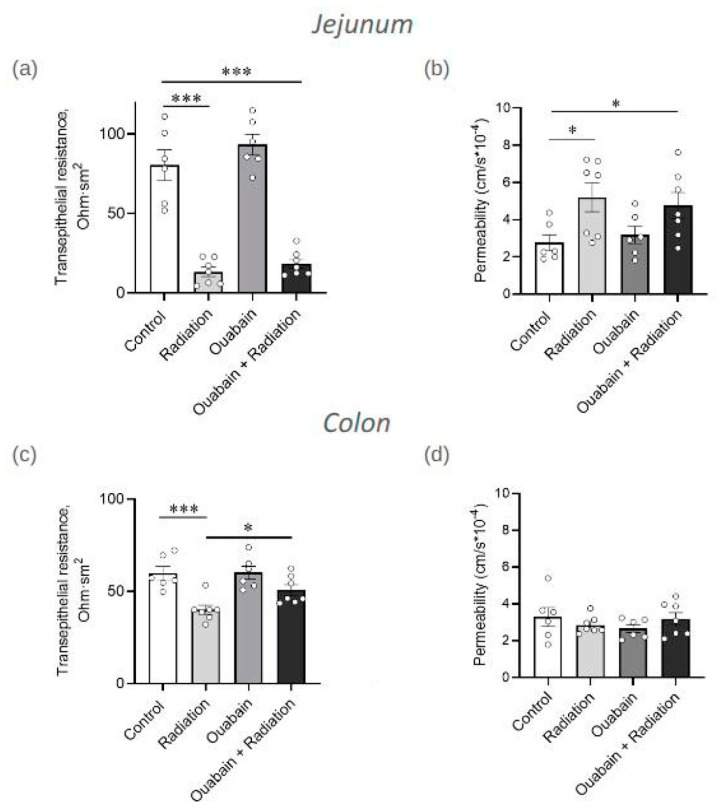
Transepithelial resistance (TER) and paracellular permeability of jejunum (**a,b**) and colon (**c**,**d**) tissues measured in an Ussing chamber. The measurements of TER (**a,c**) were carried out after 15 min of incubation in an Ussing chamber. Paracellular permeability (**b,d**) was measured as the paracellular flux of sodium fluorescein (376 Da). The number of symbols corresponds to the number of animals. For each rat, 3–4 segments of tissue were examined. The one-way ANOVA was followed by the Bonferroni multiple comparisons test. * *p* < 0.05, *** *p* < 0.001—compared as indicated by horizontal bars.

**Figure 3 ijms-25-00278-f003:**
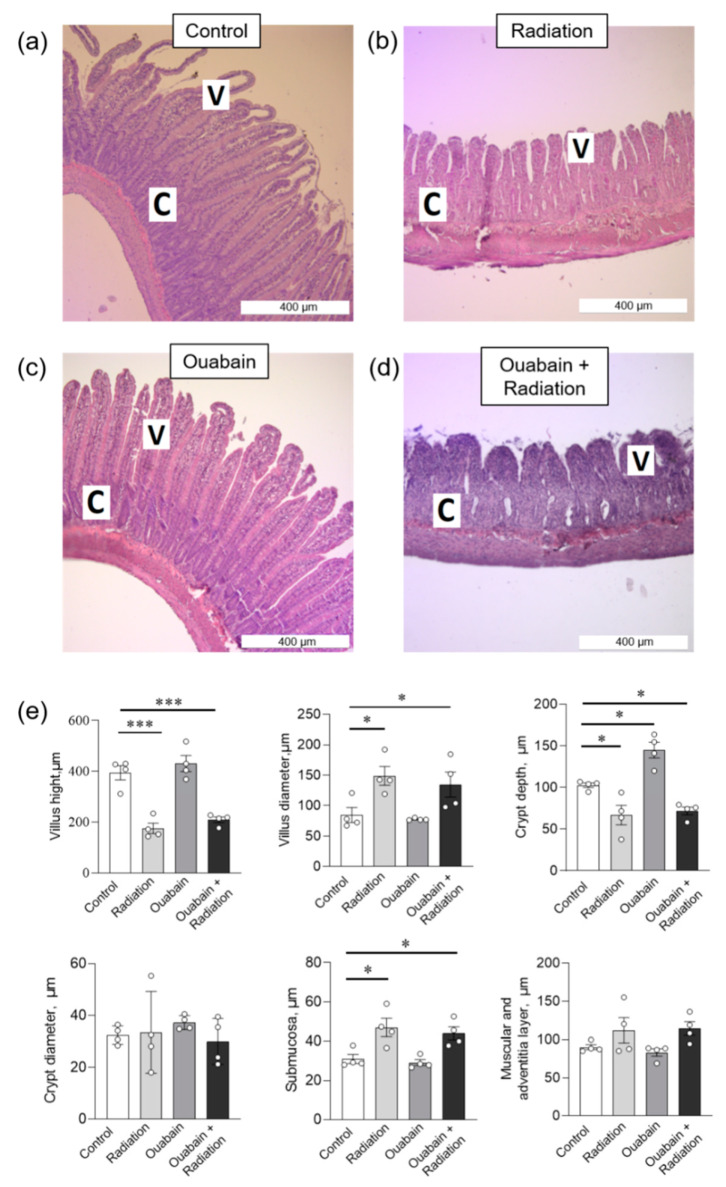
Histomorphometric parameters of the rat jejunum tissue. (**a**–**d**) H&E (hematoxylin-eosin) stained sections of jejunum; C—crypt, V—villus. (**e**) Morphometric parameters. The number of symbols corresponds to the number of animals. The one-way ANOVA was followed by the Bonferroni multiple comparisons test. * *p* < 0.05, *** *p* < 0.001—compared as indicated by horizontal bars.

**Figure 4 ijms-25-00278-f004:**
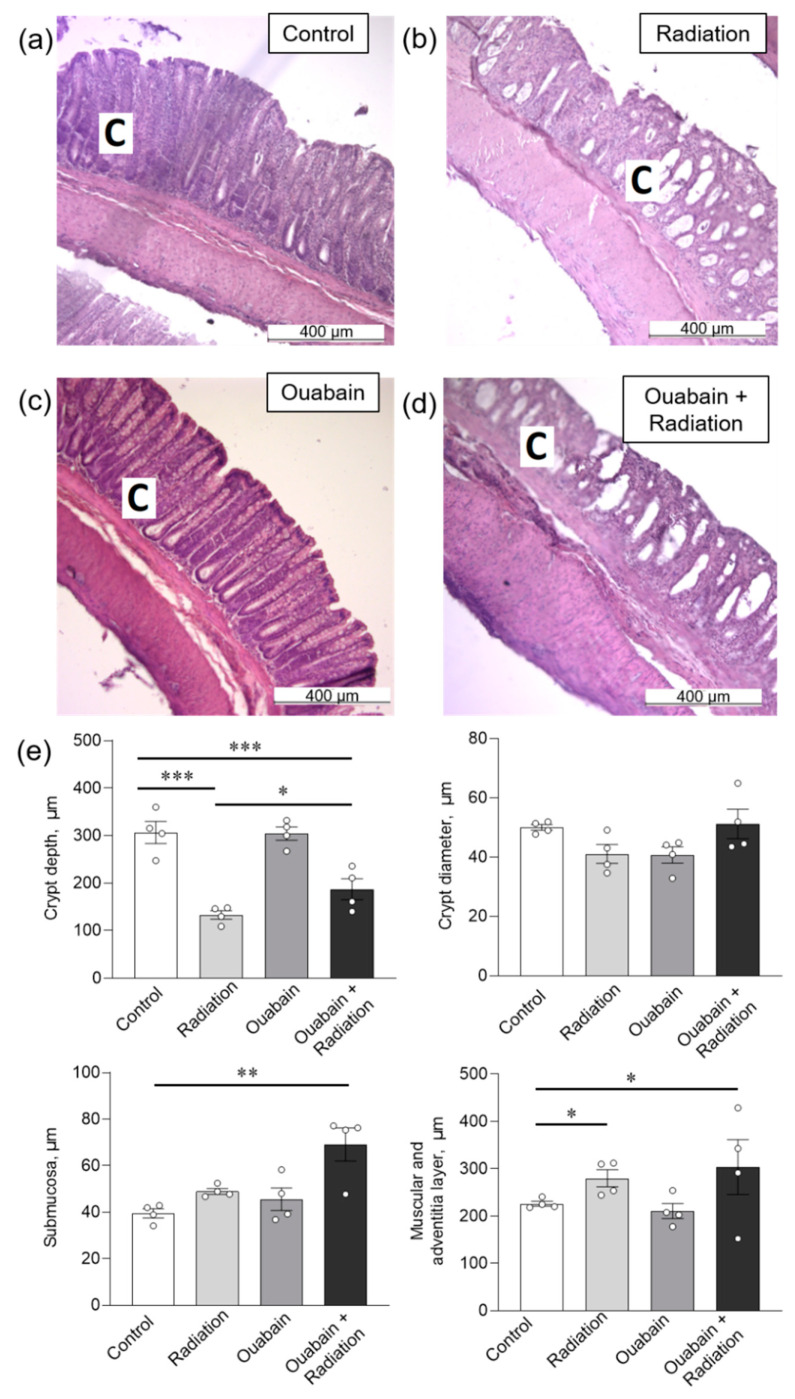
Histomorphometric parameters of the rat colon tissue. (**a**–**d**) H&E stained sections of the colon; C—crypt. (**e**) Morphometric parameters. The number of symbols corresponds to the number of animals. The one-way ANOVA was followed by the Bonferroni multiple comparisons test * *p* < 0.05, ** *p* < 0.01, *** *p* < 0.001—compared as indicated by horizontal bars.

**Figure 5 ijms-25-00278-f005:**
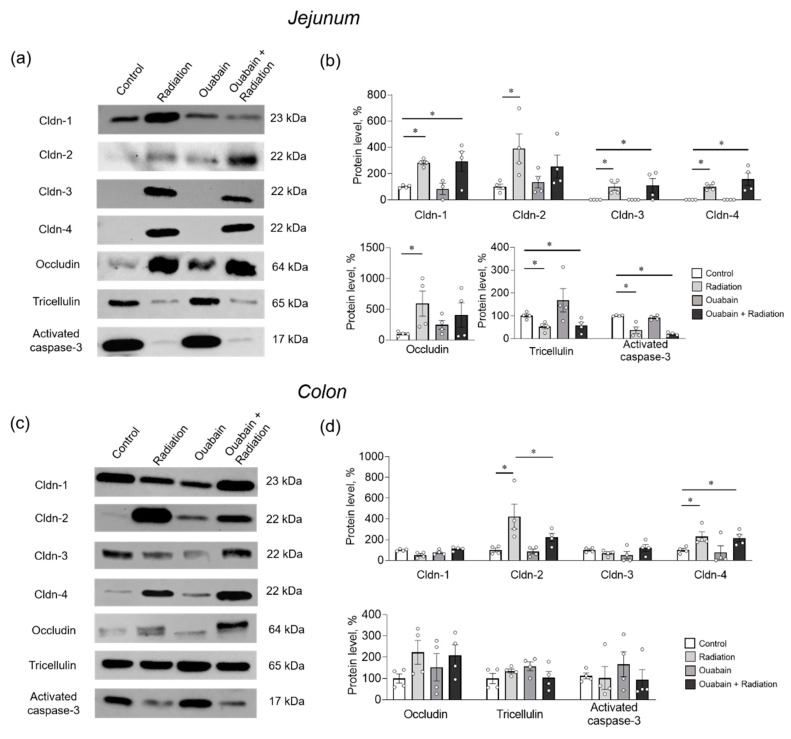
Level of TJ proteins and activated caspase-3 in the rat jejunum (**a,b**) and colon (**c,d**) tissues. (**a,c**) Representative Western blots. Total protein was used to normalize signal densities (see Appendix A). (**b,d**) Stain Free^TM^ Western blot analysis of protein levels. The number of symbols corresponds to the number of animals. The one-way ANOVA was followed by the Bonferroni multiple comparisons test. * *p* < 0.05—compared as indicated by horizontal bars.

**Figure 6 ijms-25-00278-f006:**
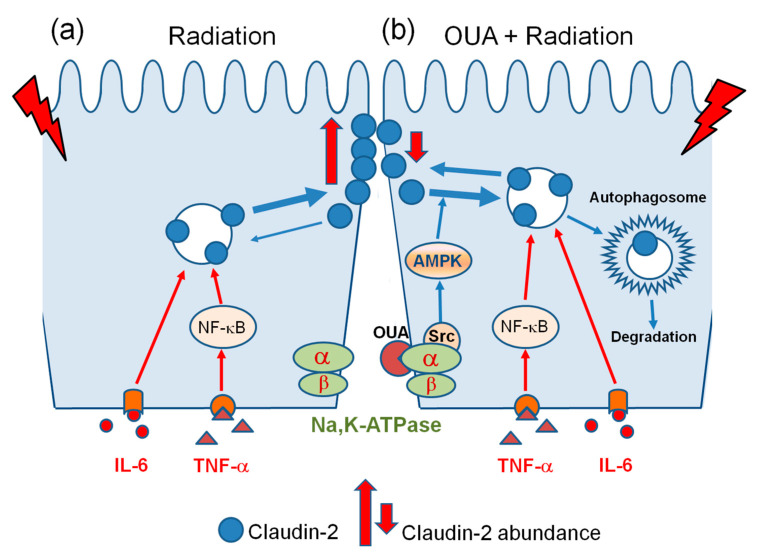
Schematic representation of the regulation of claudin-2 abundance by relatively low (**a**) and high (**b**) concentrations of circulating ouabain, 2.5 nM and 17.3 nM, respectively, as measured in this study (see text).

## Data Availability

The data that support the findings of this study are available from the corresponding author upon reasonable request.

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
