# Peer review of "Chronic Ouabain Targets Pore-Forming Claudin-2 and Ameliorates Radiation-Induced Damage to the Rat Intestinal Tissue Barrier"

_ijms, 2023, doi:10.3390/ijms25010278_

Round 1

Reviewer 1 Report

Comments and Suggestions for Authors

Major comments:

1.     What kind of environment that people will be exposed in “10Gy X-ray irradiation”. Please describe in Introduction.

2.     Fig 1 reveals that IR enhances Quabain keep in blood. What is the mechanism to keep Quabain in blood by IR.

3.     Quabain may prevent IR-mediated colon dysfunctions and Cldn-2 expression, please demonstrate that Cldn-2 is specifically associated with colon damage.

4.     What is the mechanism that IR alters the gene expression  in Jejunum and colon.

Author Response

Reviewer 1

Major comments:

The authors are grateful to the referee for a detailed acquaintance with their work and valuable comments.

  1. What kind of environment that people will be exposed in “10Gy X-ray irradiation”. Please describe in Introduction.

Intraoperative radiation therapy dose generally being in the range of 10–20 Gy [Girinsky et al., 1994; Haddock, 2017]. Similar dose load is impacted on personal by nuclear power plant incidents and radiation accidents, for example Chernobyl nuclear plant workers received doses up to 16 Gy [Bouville, Kryuchkov, 2014].

We provided these clarifications in lines 35-38.

  1. Fig 1 reveals that IR enhances Quabain keep in blood. What is the mechanism to keep Quabain in blood by IR.

The mechanisms of these changes in serum ouabain concentrations are unclear. First, the form in which ouabain circulates is poorly understood. It is proposed that ouabain is transported in complexes with carrier(s) proteins that provide a reservoir/buffer for ouabain and protection from metabolism and renal clearance. It is also impossible to exclude some feedback mechanisms that regulate the dissociation of ouabain from the carrier and, consequently, the level of its circulation [Parhami-Seren et al., 2002]. In this study, six days of exogenous ouabain administration increased serum ouabain levels. Studies in dogs and humans have shown that following a single intravenous injection, plasma ouabain concentration, after an initial rapid decline, further decline slowly with a half-life of 18–24 hours. Repeated daily administration of ouabain to humans resulted in plateau plasma concentrations after 4–5 days, suggesting equilibrium between ouabain accumulation and clearance [Selden, Smith, 1972]. How IR may influence these mechanisms has not been studied. Secondly, the synthesis and release of endogenous ouabain is regulated by the sympathetic nervous system [Blaustein, Hamlyn, 2020; Leenen et al., 2020], and the hypothalamic-pituitary-adrenal axis is known to be activated by IR [Lebaron-Jacobs et al., 2004]. Thus, an increase in ouabain levels in rats receiving exogenous ouabain and exposed to IR may further activate the sympathetic nervous system, which may additionally stimulate the release of ouabain from the adrenal glands.

This issue is now addressed in lines 269-285.

  1. Quabain may prevent IR-mediated colon dysfunctions and Cldn-2 expression, please demonstrate that Cldn-2 is specifically associated with colon damage.

The role of claudin-2 in the colon depends on the agent or conditions that caused its upregulation. In the case of bacterial infection or toxins, an increase in claudin-2 levels is considered an adaptive process aimed at flushing out the toxin or bacteria (for review [Horowitz et al., 2023]. In inflammation of the colon, such as Crohn's disease, an increase in claudin-2 is considered to be a pathogenetic change in the state of the intestinal barrier [Zeissig et al., 2007].

We provided these clarifications in lines 290-297.

  1. What is the mechanism that IR alters the gene expression in Jejunum and colon.

The following parts have been added to address this issue:

The mechanisms by which radiation affects gene expression are complex and multifaceted. IR directly interacts with DNA molecules, causing various types of DNA damage, such as single-strand breaks, double-strand breaks, and base damage [Ward, 1988; Nakano et al., 2022].

Lines 206-208.

Genes involved in antioxidant defense mechanisms, such as superoxide dismutase and catalase, may be induced in response to radiation-induced oxidative stress [Xiao et al., 2015].

Lines 210-212.

Reviewer 2 Report

Comments and Suggestions for Authors

            The present studies investigate a potential for mitigation of damage from irradiation in the large intestine.  A strength and interesting, important aspect is that the jejunal segment of the small intestine undergoes similar damage from irradiation as the colon, but providing ‘chronic’ ouabain does not protect against the damage.  Irradiation induced damages appear at least in part due to changes in the pore forming claudin2 expression and presumably function, as well as potentially due to occludin changes which may regulate claudin 2 activity but has other functions.  In consideration of the complex nature of the injury and ouabain protection, a graphical presentation of this is needed.  The text so far is incomplete, it asks the reader to interpret too much.   There is speculation by the authors, but it is not presented clearly. 

            There are a number of greater and pivotal importance for reviewing this study.  The authors have been positive and claim that the endogenous ouabain could protect against irradiation damage of the colon.  Please read the end of the abstract and many sections of the Discussion. A picture is worth 1000 words as the expression goes.  Histology is always the gold standard to assess damage and in Figure 4 the appearance of the colon tissue with radiation and ouabain appears worse than the irradiation alone in this reviewer’s assessment.  I read the crypt length and number data and the section shown may not be representative.  Serum citrulline has been used for small intestinal damage after irradiation and it is not known about colonic damage.  Indeed this study of colonic damage after radiation is novel, over 25 publications from a Google search on irradiation intestinal damage were checked and all were on small intestinal damage.  That being said the histology of colonic damage is telling and not consistent with less damage. The authors present no damage on health and recovery from sublethal radiation, this data should be shown.  This was not designed or presented as a survival study it is appreciated, but the submucosal edema is worse by appearance in ouabain with irradiation.  

            This leads to the second related point is that the transepithelial permeability.  The in vitro Ussing chamber has used a FITC-dextran which is acceptable, but the histology damage must be resolved with the lack of change of permeability.  The dextran could be too large and therefore not reflect changes, a small marker such as mannitol (180Daltons) should also be presented.  One observes large areas of clearness and what this is I cannot tell from the images, but it is not preserved or protected colon epithelium and therefore preserved and restrictive permeability. Regarding the other data,  I would not state the crypt depth is much different than irradiation nor crypt architecture, it appears damaged to me.  The studies need to assess paracellular permeability which is indeed influenced by claudin 2 by permeability markers.  One cannot do whole animal FITC dextran studies as one cannot distinguish longitudinal segments, eg jejunum versus colon, you get the whole intestine.   However, ex vivo colonic loops and use of both a smaller marker such as mannitol (180Da) and a FITC-dextran (2500Da) need to be done.  It is suspected by this reviewer increases in permeability will be observed not prevented by ouabain, inconsistent with the present author hypothesis. The manuscript needs to explain how the permeability changes are preserved. Additionally,  Data on ouabain and short circuit current does not make sense.  Ouabain, at least as higher concentrations, nearly completely blocks the NaKATPase and this will eliminate both absorptive or secretory movement mucosal to serosal or vice versa. Only paracellular movement would remain and this should not generate Isc!  This finding may be very concentration dependent and needs explanation.  It needs to be made clear that increased permeability is not beneficial, preservation of restricted solute flow across the colon is normal function so that the colon may concentrate the stool.  How claudin 2, occluding, the other claudins or other proteins such as JAM participate is only partially understood.  If the NaKATPase participates in the paracellular regulation in the colon would be of interest and have importance. 

            Of less importance, greater background is needed on the endogenous ouabain.  How does its structure resemble and different from ouabain?  Every publication on an agent should allow the reader to have sufficient background and this research group has been one of only a small number to investigate this important endogenous compound.  This background does not  need to be extensive, 3-5 sentences will suffice and that will allow greater appreciation for the present results without needing to read additional publications (which an interested reader will certainly do). 

            To enhance the manuscript and potential importance of the studies, use for therapy tough, digoxin is used uncommonly due to the narrow therapeutic window.  Also the use in humans may be different as sensitivity to ouabain or digoxin is lower in rodents, they may tolerate higher doses and this could effect in vivo use. Until data is available on the question of sensitivity of the alpha1 isoform of human versus mouse/rat relating to toxicity and sensitivity, this should be considered. The question of sensitivity and action also impinges on a separate concern.  Could some of the effects of ouabain or endogenous ouabain on physiological actions of the ATPase, eg cytoskeletal regulation and association versus activity to maintain low Na, occur at different concentrations?  One might speculate that one activity requires higher concentrations and this is important to consider.  s

            Also, data on ouabain and short circuit current does not make sense.  Ouabain, at least as higher concentrations, nearly completely blocks the NaKATPase and this will eliminate both absorptive or secretory movement mucosal to serosal or vice versa. Only paracellular movement would remain and this should not generate Isc!  This finding may be very concentration dependent and needs explanation. 

            In conclusion, from what I can analyze, the data on ouabain protection against colonic irradiation damage is not as strong as concluded.  If the authors can present a model which explains their data better and argues for protection and then future potential therapeutic use, I would encourage and look forward to the revision.

Comments on the Quality of English Language

English usage acceptable

Author Response

Reviewer 2

Comments and Suggestions for Authors

 1.The present studies investigate a potential for mitigation of damage from irradiation in the large intestine. A strength and interesting, important aspect is that the jejunal segment of the small intestine undergoes similar damage from irradiation as the colon, but providing ‘chronic’ ouabain does not protect against the damage. Irradiation induced damages appear at least in part due to changes in the pore forming claudin2 expression and presumably function, as well as potentially due to occludin changes which may regulate claudin 2 activity but has other functions. In consideration of the complex nature of the injury and ouabain protection, a graphical presentation of this is needed. The text so far is incomplete, it asks the reader to interpret too much. There is speculation by the authors, but it is not presented clearly.

The authors are grateful to the referee for a detailed acquaintance with their work and valuable comments.

A graphical presentation and corresponding explanation are added (line 302-333).

IR induces the overproduction of ROS, which is a multifaceted regulator that plays an important role in various pathways involved in maintaining cellular homeostasis and regulating key transcription factors [Checa and Aran, 2020]. The factors that mediate claudin-2 upregulation are not fully characterized, but various studies have shown that tumor necrosis factor (TNF) and a number of interleukins are potential enhancers of claudin-2 expression [Horowitz et al., 2023]. IR is known to enhance the production of TNF-α and interleukins [Yahyapour et al., 2018; Banerjee et al., 2019]. In particular, total body irradiation at a dose of 10 Gy increases serum levels of TNF-α and IL-6 in humans [Girinksy ea 1994], as well as in human endothelial cells [Meeren et al., 1997]. It has been suggested that TNFα-dependent upregulation of claudin-2 may occur through the phosphatidylinositol-3-kinase pathway [Mankertz et al., 2009]. The role of TNF-α/NF-kB signaling in the upregulation of claudin-2 has also been reported [Amasheh et al., 2010; Ahmad et al., 2017]. The activity of claudin-2 promoter has been shown to be increased by IL-6 in a MEK/ERK and PI3K dependent manner, resulting in increased expression of claudin-2 [Suzuki et al., 2011].

It is noteworthy that IR (8.5 Gy) at 3.5 days after irradiation changed the localization of claudin-2 in epithelial cells of the mice small intestine [Banerjee et al., 2019]. It has also been shown that increased expression of claudin-2 is associated with increased binding of claudin-2 and caveolin-1 [Ares et al., 2019]. Caveolin-1 is a vital protein for many cellular processes, including autophagy [Dalton et al., 2023]. Thus, increased binding of caveolin-1 and claudin-2 suggests that caveolin-1 may act as a shuttle mechanism for increased internalization of claudin-2 [Ares et al., 2019].

Notably, ouabain is able to activate AMP-activated protein kinase (AMPK), a master regulator of cellular metabolism in eukaryotes that plays a critical role in cellular processes such as autophagy [Wang et al., 2022]. Moreover, ouabain at a concentration of 25 nM (which is similar to the 17.3 nM observed in this study) activates AMPK and positively regulates autophagy in human cancer cell lines [Shen et al., 2020]. Thus, this autophagic degradation pathway triggered by ouabain may further enhance caveolin-1-dependent claudin-2 internalization. If so, such a mechanism could explain the preventive effect of ouabain administration against IR-induced claudin-2 upregulation.

Combining the literature and our findings, we propose the following hypothetical scheme, presented in Figure 6.

2.There are a number of greater and pivotal importance for reviewing this study. The authors have been positive and claim that the endogenous ouabain could protect against irradiation damage of the colon. Please read the end of the abstract and many sections of the Discussion. A picture is worth 1000 words as the expression goes. Histology is always the gold standard to assess damage and in Figure 4 the appearance of the colon tissue with radiation and ouabain appears worse than the irradiation alone in this reviewer’s assessment. I read the crypt length and number data and the section shown may not be representative. Serum citrulline has been used for small intestinal damage after irradiation and it is not known about colonic damage. Indeed this study of colonic damage after radiation is novel, over 25 publications from a Google search on irradiation intestinal damage were checked and all were on small intestinal damage. That being said the histology of colonic damage is telling and not consistent with less damage. The authors present no damage on health and recovery from sublethal radiation, this data should be shown. This was not designed or presented as a survival study it is appreciated, but the submucosal edema is worse by appearance in ouabain with irradiation.

We agree with the reviewer's comments. The text has been amended accordingly:

In the colon tissue, ouabain prevented electrophysiological disturbances and claudin-2 overexpression. Lines 21-23.

Chronic ouabain by itself did not affect any parameters, could not prevent the observed disturbances (Figure 4).

Lines 175-177.

Experiments with different doses of total-body X-ray IR (nonlethal — 2 Gy, half-lethal — 5 Gy, and lethal — 10 Gy ranges) showed that only exposure to a dose of 10 Gy led to the manifestation of physiological effects on the barrier functions of the rat jejunum and colon 72 h after IR

Lines 60-63.

3.This leads to the second related point is that the transepithelial permeability. The in vitro Ussing chamber has used a FITC-dextran which is acceptable, but the histology damage must be resolved with the lack of change of permeability. The dextran could be too large and therefore not reflect changes, a small marker such as mannitol (180Daltons) should also be presented. One observes large areas of clearness and what this is I cannot tell from the images, but it is not preserved or protected colon epithelium and therefore preserved and restrictive permeability. Regarding the other data, I would not state the crypt depth is much different than irradiation nor crypt architecture, it appears damaged to me. The studies need to assess paracellular permeability which is indeed influenced by claudin 2 by permeability markers. One cannot do whole animal FITC dextran studies as one cannot distinguish longitudinal segments, eg jejunum versus colon, you get the whole intestine. However, ex vivo colonic loops and use of both a smaller marker such as mannitol (180Da) and a FITC-dextran (2500Da) need to be done. It is suspected by this reviewer increases in permeability will be observed not prevented by ouabain, inconsistent with the present author hypothesis. The manuscript needs to explain how the permeability changes are preserved. Additionally, Data on ouabain and short circuit current does not make sense. Ouabain, at least as higher concentrations, nearly completely blocks the NaKATPase and this will eliminate both absorptive or secretory movement mucosal to serosal or vice versa. Only paracellular movement would remain and this should not generate Isc! This finding may be very concentration dependent and needs explanation. It needs to be made clear that increased permeability is not beneficial, preservation of restricted solute flow across the colon is normal function so that the colon may concentrate the stool. How claudin 2, occluding, the other claudins or other proteins such as JAM participate is only partially understood. If the NaKATPase participates in the paracellular regulation in the colon would be of interest and have importance. 

We agree with the reviewers regarding short-circuit current and remove this data from all sections of the manuscript.

The following clarifications were made:

Paracellular permeability measured as the paracellular flux of sodium fluorescein (376 Da).

Line 151.

Paracellular permeability was measured in an Ussing chamber as sodium fluorescein (376 Da) flux.

Line 156.

50 μl of 376 Da sodium fluorescein (Sigma Aldrich, Burlington, MA, USA)

Line 413.

4.Of less importance, greater background is needed on the endogenous ouabain. How does its structure resemble and different from ouabain? Every publication on an agent should allow the reader to have sufficient background and this research group has been one of only a small number to investigate this important endogenous compound. This background does not need to be extensive, 3-5 sentences will suffice and that will allow greater appreciation for the present results without needing to read additional publications (which an interested reader will certainly do). 

Na,K-ATPase is a ubiquitous transport protein that pumps Na+ out of the cell and K+ in the opposite direction across the plasma membrane. The catalytic and transport α subunit of Na,K-ATPase is expressed in four isoforms in a cell- and tissue-specific manner. Most cells co-express the α1 isoform in combination with other α isoforms, whereas red blood cells, kidneys, lungs, and intestine predominantly express the α1 isoform [Blanco, Mercer, 1998; Clausen et al., 2017]. The extracellular loops of α subunit form a highly specific binding site for cardiotonic steroids such as ouabain and digoxin, and this site is the only known receptor for these ligands [Lingrel, 2010]. Ouabain is a cardiotonic steroid type compound. Ouabain, originally extracted from plants (e.g., Strophanthus gratus and Acokanthera ouabaio), has an endogenous circulating analog that has been isolated from mammalian tissues and fluids and is structurally, biochemically, and immunologically indistinguishable from exogenous ouabain [Schoner, 2002]. Ouabain is believed to be synthesized in the adrenal gland cortex and hypothalamus, and is currently suggested to act as a hormone [Blaustein, Hamlyn, 2020].

This issue is addressed now in lines 221-233.

5.To enhance the manuscript and potential importance of the studies, use for therapy tough, digoxin is used uncommonly due to the narrow therapeutic window. Also the use in humans may be different as sensitivity to ouabain or digoxin is lower in rodents, they may tolerate higher doses and this could effect in vivo use. Until data is available on the question of sensitivity of the alpha1 isoform of human versus mouse/rat relating to toxicity and sensitivity, this should be considered. The question of sensitivity and action also impinges on a separate concern. Could some of the effects of ouabain or endogenous ouabain on physiological actions of the ATPase, eg cytoskeletal regulation and association versus activity to maintain low Na, occur at different concentrations? One might speculate that one activity requires higher concentrations and this is important to consider.

Relevant explanations added:

However, it is important to note that, unlike in rodents, the α1 isozyme in rabbit, pig, dog, sheep, guinea pig, and human is relatively sensitive to ouabain. In human, a similar affinity of all α subunit isozymes with ouabain binding constants in the nanomolar concentration range has been shown [Blanco, Mercer, 1998; Lingrel, 2010; Katz et al., 2010; Cherniavsky Lev et al., 2015]. This leaves open the question of how far the results of our study can be extrapolated to subjects other than rats, and this may be considered a limitation of this work.

Lines 349-354.

Importantly, the cellular cytoskeleton, composed by a dynamic network of microfilaments, intermediate filaments and microtubules, plays an important role in cell polarity, barrier integrity, intracellular trafficking and intestinal epithelium homeostasis [Perrin & Matic Vignjevic, 2023]. The ROCK (Rho-associated coiled-coil forming kinase) signaling pathway is considered a key regulator of the cytoskeleton components [Guan et al., 2023]. In cultured epithelial (MDCK) cells, ouabain (10 nM) has been shown to induce transcript changes and activation of ROCK signaling [Martínez-Rendón et al., 2023], and this finding opens a new field of further research on claudin-2 regulation. Whether occludin, which is associated with the actin cytoskeleton and regulated by the ROCK signaling pathway [Feng et al., 2018], may be involved in the regulation of claudin-2 expression triggered by ouabain remains to be determined.

Lines 334-343.

It should be noted that in rodents, the α1 Na,K-ATPase isozyme is highly resistant to ouabain (IC50 values range from tens to hundreds micromolar), compared to other isozymes that is two to four orders of magnitude more sensitive [Blanco, Mercer, 1998; Lingrel, 2010]. This suggests that the circulating ouabain concentrations in our study trigger α1Na,K-ATPase/Src-dependent intracellular signaling rather than inhibiting enzyme activity and altering Na+ balance.

Lines 344- 348.

6.In conclusion, from what I can analyze, the data on ouabain protection against colonic irradiation damage is not as strong as concluded. If the authors can present a model which explains their data better and argues for protection and then future potential therapeutic use, I would encourage and look forward to the revision.

We agree with the Reviewer and now use more careful language:

These observations may reveal a mechanism by which circulating ouabain maintain tight junctions integrity under IR-induced intestinal dysfunction.

Lines 25-26.

Altogether, our new findings corroborate a functional α1Na,K-ATPase/claudin interaction and the importance of ouabain as a circulating regulator of this interaction that can modulate claudin-2 expression. Further studies of dose-, time-, and use-dependence are needed to more accurately evaluate the therapeutic potential of ouabain treatment for IR-induced intestinal injury.

Lines 354-358.

Reviewer 3 Report

Comments and Suggestions for Authors

Comments to the Authors of manuscript number: biomedicines-2739231 entitled “Chronic Ouabain Targets Pore-Forming Claudin-2 and Ameliorates Radiation-Induced Damage to the Rat Intestinal Tissue Barrier”.

This study investigated the impact of ionizing radiation (IR) on the gastrointestinal tract and explored the potential therapeutic effects of ouabain, a specific ligand of Na,K-ATPase. Male Wistar rats received intraperitoneal injections of ouabain or vehicle for six days, with IR exposure on the fourth day. The results showed that ouabain had protective effects on certain electrophysiological and molecular aspects in the jejunum and colon tissues, specifically targeting claudin-2. These findings suggest a novel approach to mitigating IR-induced intestinal dysfunction.

I have carefully reviewed that manuscript and regret to inform that it cannot be accepted for publication. Several crucial elements are missing in the paper, making it challenging to justify the acceptance of this paper. There is no detailed explanation or verification of the radiation doses used in the study. Given the potentially harmful effects of radiation exposure, it is imperative to provide a robust rationale for the chosen doses and to verify their appropriateness. Lack of such information raises concerns about the validity and safety of the experimental design.

1. L 34- it is good to write typical IR in cancer therapy or what is IR during space flights. In this case this mentioned value does not indicate nothing

2. L 57 – reference needed

3. L 59- earlier only claudin were described, add occluding and other

4. L 60- the changes should be described clearly

5. L 63- what does it mean:”different sensitivity”?

6. L 63 – this IR is typical in what cases?

7. L 64- levels?

8. L 70 -it determines secondary active transport

9. There is no hypothesis

10. there is no aim of the study. The manuscript fails to adequately articulate the necessity of conducting this particular research. It is essential to clearly establish why this model is needed and what specific gaps or issues in the current understanding it aims to address. Without a compelling rationale, the significance of the study is unclear.

11. Figure 3. These are not histopathological parameters. These are normal typical histomorphological parameters. There is a difference between pathology and correct histology

12. Figure 4. The same as above

13. L 286 – the dose of IR is unclear. How this dose was determined.

14. the study is prepared not correctly. It is not know if this dose is considered "correct", because there is no specific goals of the study. Different studies may use different radiation doses based on the intended experimental outcomes. In radiation research, it's essential to carefully select doses to achieve the desired effects without causing excessive harm to the subjects. The appropriateness of the dose should be evaluated based on the study objectives, relevant literature, and ethical considerations, and preliminary study.

15. The manuscript lacks discussion on how the findings of this study may translate into advancements or insights in related fields. Providing a broader context and discussing potential implications in other relevant disciplines should be presented.

16. L 286- shame IR – totally unclear

17. 164 – V?

18. 318- patomorphology?

19. Conducting research on animals solely for the sake of research raises ethical concerns. The manuscript should explicitly address the ethical considerations involved in using animal subjects and demonstrate a clear and justifiable reason for the chosen model.

Author Response

Reviewer 3

Comments and Suggestions for Authors

 This study investigated the impact of ionizing radiation (IR) on the gastrointestinal tract and explored the potential therapeutic effects of ouabain, a specific ligand of Na,K-ATPase. Male Wistar rats received intraperitoneal injections of ouabain or vehicle for six days, with IR exposure on the fourth day. The results showed that ouabain had protective effects on certain electrophysiological and molecular aspects in the jejunum and colon tissues, specifically targeting claudin-2. These findings suggest a novel approach to mitigating IR-induced intestinal dysfunction.

I have carefully reviewed that manuscript and regret to inform that it cannot be accepted for publication. Several crucial elements are missing in the paper, making it challenging to justify the acceptance of this paper. There is no detailed explanation or verification of the radiation doses used in the study. Given the potentially harmful effects of radiation exposure, it is imperative to provide a robust rationale for the chosen doses and to verify their appropriateness. Lack of such information raises concerns about the validity and safety of the experimental design.

The authors are grateful to the referee for a detailed acquaintance with their work and valuable comments.

The following part has been added to the Introduction:

Experiments with different doses of total-body X-ray IR (nonlethal — 2 Gy, half-lethal — 5 Gy, and lethal — 10 Gy ranges) showed that only exposure to a dose of 10 Gy led to the manifestation of physiological effects on the barrier functions of the rat jejunum and colon 72 h after IR

Lines 60-63.

L 34- it is good to write typical IR in cancer therapy or what is IR during space flights. In this case this mentioned value does not indicate nothing

This information is presented in the Introduction.

…during space flights (0.3–0.6 mGy/day) [4]. Intraoperative radiation therapy dose generally being in the range of 10–20 Gy. Similar dose load is impacted on personal by nuclear power plant incidents and radiation accidents, for example Chernobyl nuclear plant workers received doses up to 16 Gy.

Lines 34-38.

  1. L 57 – reference needed

Link to articles included. Line 64.

  1. L 59- earlier only claudin were described, add occluding and other

This issue is addressed in lines 55-59.

  1. L 60- the changes should be described clearly

This information is added:

After IR at dose lower than range causing GIARS, decrease of occludin, claudin-3 and ZO-1 was shown in murine ileum and colon [Shukla et al., 2016].

Lines 66-67.

  1. L 63- what does it mean: ”different sensitivity”?

The following clarifications was made:

This means that electrophysiological parameters, permeability, changes in histological structure, and expression of TJ proteins in the colon and small intestine differ when ex-posed to the same dose of 10 Gy.

Lines 70-72.

  1. L 63 – this IR is typical in what cases?

Authors address this question in the Lines 34-38 and 60-63.

  1. L 64- levels?

The term «level» has been replaced by «expression»

Lines 70-72

  1. L 70 -it determines secondary active transport

Secondary active transport is based on primary active transport, in which the main role belongs to Na,K-ATPase.

  1. There is no hypothesis

Please see below, paragraph 10

  1. There is no aim of the study. The manuscript fails to adequately articulate the necessity of conducting this particular research. It is essential to clearly establish why this model is needed and what specific gaps or issues in the current understanding it aims to address. Without a compelling rationale, the significance of the study is unclear.

We have changed the final paragraph of the introduction, formulating more precisely the hypothesis and, therefore, the aim of the study:

Previously, in animal models of various pathological conditions, injections of ouabain in doses of 1–1.8 μg/kg demonstrated their effectiveness [Dvela-Levitt et al., 2014; Garcia et al., 2019; Markov et al., 2020; Kravtsova et al., 2022]. In summary, we hypothesized that elevations of circulating ouabain by its exogenous administration could maintain intestinal barrier function during IR-induced injury. We tested this hypothesis in rats intraperitoneally injected with ouabain in a dose of 1 μg/kg and consequently exposed to total-body X-ray IR (10 Gy). In isolated tissues of the jejunum and colon, electrophysiological characteristics and paracellular permeability were studied; Western blotting and histological analysis were also performed.

Lines 104-111

  1. Figure 3. These are not histopathological parameters. These are normal typical histomorphological parameters. There is a difference between pathology and correct histology

The text has been amended accordingly. Changed to morphometric parameters.

Lines 170.

  1. Figure 4. The same as above

The text has been amended accordingly.

Line 179.

  1. L 286 – the dose of IR is unclear. How this dose was determined.

The following clarifications was made:

The RUM-17 unit emits X-rays with a dose rate of 0.31 Gy/min. This means that if an animal, fixed in a plexiglass box, is under the switched on X-ray tube for a minute, it receives an absorbed dose of 0.31 Gy. Accordingly, after 31 minutes it receives 10 Gy. To check the absorbed dose, an individual dosimeter was used, followed by result interpretation with a GO-32 measuring device (Spetsoborona, Russia).

Lines 390-393.

If the question concerns the choice of a specific dose (10 Gy), then this choice was carefully justified in the Introduction (added parts, lines 34-38 and 60-63)

  1. the study is prepared not correctly. It is not know if this dose is considered "correct", because there is no specific goals of the study. Different studies may use different radiation doses based on the intended experimental outcomes. In radiation research, it's essential to carefully select doses to achieve the desired effects without causing excessive harm to the subjects. The appropriateness of the dose should be evaluated based on the study objectives, relevant literature, and ethical considerations, and preliminary study.

The dose was chosen based on ethical considerations, the purpose of the study, literature data and our own preliminary experiments. This information is presented in the Introduction.

  1. The manuscript lacks discussion on how the findings of this study may translate into advancements or insights in related fields. Providing a broader context and discussing potential implications in other relevant disciplines should be presented.

We have significantly changed the content of the discussion, presented a graphical representation of the hypothesis and examined the results from a broader perspective.

  1. L 286- shame IR – totally unclear

The term “sham” IR is used throughout the text.

Sham irradiation means that the animals were placed into X-ray unit with the radiation X-ray tube turned off, in a manner similar to animals receiving a dose of 10 Gy. Thus, sham exposure is a control for the irradiation procedure.

Lines 393-395.

  1. 164 – V?

Villus was deleted from the Figure 4 legend.

  1. 318- patomorphology?

We believe that «Histological Analysis» objectively reflects this method.

Line 426.

  1. Conducting research on animals solely for the sake of research raises ethical concerns. The manuscript should explicitly address the ethical considerations involved in using animal subjects and demonstrate a clear and justifiable reason for the chosen model.

Experiments were conducted to study the molecular mechanisms of preventing intestinal damage from radiation, which required the use of an animal study model. Ethical review was carried out by the ethics committee, as stated in Institutional Review Board Statement.

Lines 482-485.

Round 2

Reviewer 1 Report

Comments and Suggestions for Authors

No more questions.

Reviewer 2 Report

Comments and Suggestions for Authors

The revision conscientiously addresses all of my comments.

Reviewer 3 Report

Comments and Suggestions for Authors

I have no comments